# UniSparse: Combining Weight Pruning and Spike Sparsification in Spiking Neural Networks

**Xinyu Shi** [1 2]  **Tong Bu** [1 2]  **Zhaofei Yu** [1 2]

## Abstract

Spiking Neural Networks (SNNs) offer a notable energy-saving advantage compared to Artificial Neural Networks (ANNs) when deployed on neuromorphic hardware. While recent SNNs achieve superior performance using larger and deeper backbones, this comes at a cost of diminishing their energy-saving benefits. In this paper, we propose UniSparse, a unified sparsification framework for enhancing the energy efficiency of SNNs. We demonstrate that the affine parameters in batch normalization also serve as the learnable threshold of its subsequent spiking neurons. Based on this, we propose a novel spike sparsification method that reduces firing rate by constraining the affine parameters. As a complement to spike sparsification, we propose a weight pruning method based on the same energy constraint, which can be naturally integrated with spike sparsification. Experimental results demonstrate that UniSparse achieves a state-of-the-art trade-off between accuracy and energy efficiency across models and datasets. The sparsified ResNet-18 model requires only 7.04M SOPs for inference to achieve 92.38% accuracy on the CIFAR-10 dataset. Our work highlights the great potential of deep SNNs in improving energy efficiency. Codes are available at: https://github.com/xyshi2000/UniSparse

## 1. Introduction

Spiking Neural Networks (SNNs), regarded as the third generation of Artificial Neural Networks (ANNs) (Maass, 1997), have garnered significant attention due to their notable advantages, including low power consumption, biological plausibility, and event-driven operation, which align well with neuromorphic hardware. In comparison to ANNs, SNNs exhibit superior energy efficiency due to their event-driven nature when deployed on neuromorphic platforms (Merolla et al., 2014; Pei et al., 2019).

Recent studies have increasingly focused on developing deeper, larger, and more complex SNNs to enhance performance across a range of challenging tasks, including object recognition (Hu et al., 2024; Zhou et al., 2024; Shi et al., 2024b), detection (Zhang et al., 2023; Su et al., 2023; Fan et al., 2024), segmentation (Kim et al., 2022; Yao et al., 2024; Lei et al., 2025), natural language understanding and generation (Xing et al., 2024; Zhu et al., 2024), etc. However, while these SNNs achieve impressive task performance using large-scale backbones with numerous parameters, their increased computational demands diminish their inherent energy-saving benefits. Therefore, improving the energy efficiency of deep SNN architectures has emerged as an important research direction in the field of SNNs.

Similar to ANNs, deep SNNs also exhibit considerable redundancy. The distinction, however, lies in the fact that SNNs involve two types of computational redundancy: parameter redundancy and spike redundancy. Existing approaches can be broadly categorized into two groups: model compression methods that address parameter redundancy, and spike reduction techniques that minimize unnecessary spikes (Xie et al., 2024). Unlike model compression, spike reduction is unique to SNNs and remains relatively under-explored. A key challenge is that commonly used neuron models, such as the Leaky Integrate-and-Fire (LIF) model, lack parameters or mechanisms to constrain spike firing, making it difficult to reduce spikes solely within the neuron.

Existing methods typically rely on pruning neurons using learnable masks (Shi et al., 2024a) or applying regularization on spikes to achieve spike sparsification (Deng et al., 2021), but these methods have notable limitations. For instance, the use of learnable masks introduces a large number of additional trainable parameters, significantly increasing training overhead. Moreover, it often requires carefully controlled annealing during training. On the other hand, spike regularization tends to indiscriminately suppress all spikes, which may conflict with the task optimization objective and consequently impair model performance. Therefore,

---

[1]Institute for Artificial Intelligence, Peking University [2]Beijing Key Laboratory of Brain-inspired Spiking Large Models, School of Computer Science, Peking University. Correspondence to: Zhaofei Yu <yuzf12@pku.edu.cn>.

*Proceedings of the 43rd International Conference on Machine Learning*, Seoul, South Korea. PMLR 306, 2026. Copyright 2026 by the author(s).

there is a clear need for a new method that can effectively reduce spikes without substantially compromising performance, and that can be seamlessly integrated with model compression techniques.

A straightforward approach to reduce spikes is to raise the firing threshold of spiking neurons. However, this method has limited effectiveness, as the interaction between normalization layers and spiking neurons counteracts the effect of increasing the threshold. Recently, most deep SNN structures employ normalization before spiking neuron layers, particularly batch normalization (BN), including both spiking convolutional networks (Zheng et al., 2021; Fang et al., 2021; Hu et al., 2024) and spiking vision transformers (Zhou et al., 2023; Yao et al., 2023). It helps normalize the distribution of input currents and stabilize the gradient backpropagation (Zheng et al., 2021). Beyond these advantages, we observe that the affine parameters of normalization also act as learnable thresholds for spiking neurons. While this interaction may counteract the effect of manually raising the threshold, we can leverage this characteristic in reverse, i.e., by constraining the affine parameters to reduce spikes. This spike sparsification approach offers several advantages over existing methods. First, constraining affine parameters typically eliminates relatively unimportant spikes, where membrane potentials are only slightly above the threshold, and preserves critical spikes with membrane potentials well above threshold. Second, constraining affine parameters affects only the neurons immediately following the BN layer, whereas spike regularization typically suppresses all preceding layers along with backpropagation. For layers that do not incorporate normalization, e.g., fully connected layers, since there is no interaction with affine parameters, we can directly reduce spikes by raising the firing threshold.

To complement the spike sparsification method, we propose a weight pruning method based on soft threshold reparameterization (Kusupati et al., 2020). The proposed weight pruning method derives from the same energy constraint as the spike sparsification method, enabling them to be naturally integrated into a unified framework. By combining weight pruning and spike sparsification, we propose UniSparse, a unified sparsification framework designed to enhance the energy efficiency of SNNs. The proposed UniSparse framework introduces no additional learnable parameters and controls overall sparsity through a single hyperparameter, making it both easy to use and highly adaptable to various network architectures. By reducing redundant computation while maintaining performance, UniSparse achieves a state-of-the-art balance between accuracy and energy efficiency across various models and datasets. Our main contributions are summarized as follows:

- We analyze the impact of learnable affine parameters in normalization on the neural dynamics of spiking neurons, and propose a spike sparsification method that reduces spikes by constraining the affine parameters.

- We propose a weight pruning method based on soft threshold reparameterization, which is derived from the same energy constraint as the proposed spike sparsification method, enabling them to be naturally integrated into a unified framework.

- Combining the weight pruning and spike sparsification, we propose UniSparse, a unified sparsification framework to enhance the energy efficiency of SNNs.

- We evaluate the performance of UniSparse across various models and datasets. Experimental results show that UniSparse achieves a state-of-the-art trade-off between accuracy and energy efficiency. The sparsified ResNet-18 model requires only 7.04M SOPs for inference to achieve 92.38% accuracy on CIFAR-10.

## 2. Related Work

**Pruning Methods for ANNs** To reduce redundancy in deep ANNs, numerous pruning algorithms have been proposed. Some methods leverage batch normalization (BN) layers to perform channel-level pruning. Several approaches in this category introduce sparsity-inducing regularization to the learnable scaling parameters in BN layers, such as L1 normalization (Liu et al., 2017) and polarization regularization (Zhuang et al., 2020). These regularization terms encourage the scaling factors of redundant channels to approach zero during training, thereby automatically identifying and eliminating less important channels. Some methods further refine this process by considering both the scale and shift parameters, enabling the pruning of channels that produce negative outputs (Kang & Han, 2020). In contrast, UniSparse also utilizes affine parameters to induce sparsity but does not prune channels or neurons directly. Instead, it enhances spike sparsity by constraining the distribution of input currents and suppressing redundant spiking activities. This approach is particularly well-suited for SNNs, where neurons inherently exhibit sparse firing patterns.

**Weight Pruning for SNNs** Recent research on weight pruning in SNNs has focused on adapting successful pruning methods from ANNs to SNNs. Typical approaches include magnitude-based pruning (Yin et al., 2021), alternating direction method of multipliers (ADMM) (Deng et al., 2021), and introducing learnable masks for weight pruning (Shi et al., 2024a). Furthermore, some recent studies have explored biologically inspired weight pruning methods, drawing inspiration from the similarities between SNNs and neural systems. These methods primarily focus on synaptic regrowth processes, which employ probability (Bellec et al., 2018) or gradient (Chen et al., 2021) as a criterion for

regrowth. Some methods train SNNs with sparse architectures from scratch, dynamically pruning weak connections and generating new ones according to structural plasticity rules (Shen et al., 2023). There is also work that combines the advantages of both the soft threshold reparameterization (STR) (Kusupati et al., 2020) from ANNs and the biologically inspired regrowth processes (Chen et al., 2022). Our weight pruning method is also based on soft threshold reparameterization. Different from (Chen et al., 2022), where soft thresholds rely on heuristic scheduling, we derive the update equation for the soft threshold directly from energy constraints. This enables the threshold to be automatically scheduled based on the connection patterns of each layer, and can be naturally integrated with spike sparsification.

**Reduction of Spikes**  Most existing spike reduction methods can be categorized into neuron pruning and spike sparsification. Existing neuron pruning methods either use similarity between spike trains to select neurons for pruning (Wu et al., 2019) or introduce learnable masks to prune neurons directly (Shi et al., 2024a). Current spike sparsification methods (Neil et al., 2016; Deng et al., 2021) basically achieve spike sparsification through spike regularization. These methods add a regularization term to the overall optimization objective, which indiscriminately penalizes all spikes to encourage learning of a sparse weight matrix. Recent work has also used adaptive thresholds to reduce spiking activity (Bu et al., 2026).

## 3. Preliminary

### 3.1. Spiking neuron model

In this paper, we adopt the commonly used Leaky Integrate-and-Fire (LIF) model in all spiking neural network architectures. We use the following unified discrete-time model to describe the neural dynamics of LIF neurons:

$$
\begin{aligned}
v[t] &= u[t] + \frac{1}{\tau}(x[t] - (u[t] - u_{\text{rest}})), \\
s[t] &= H(v[t] - v_{\text{th}}), \\
u[t+1] &= s[t]u_{\text{rest}} + (1 - s[t])v[t].
\end{aligned} \tag{1}
$$

Here $x[t]$ denotes the input current of the neuron at time-step $t$, $\tau$ is the membrane time constant, $u[t]$ and $v[t]$ denote the membrane potential of the neuron before and after charging at time-step $t$. $H(\cdot)$ is the Heaviside step function. $s[t] \in \{0, 1\}$ is the output of the neuron at time-step $t$, where $s[t] = 1$ denotes the neuron fires a spike. If the membrane potential after charging $v[t]$ reaches the firing threshold $v_{\text{th}}$, the neuron fires a spike, and the membrane potential after firing $u[t+1]$ will reset to the resting potential $u_{\text{rest}}$. In this paper, we set $\tau = 2$, $v_{\text{th}} = 1$, $u_{\text{rest}} = 0$.

### 3.2. Batch Normalization

Let $x$ denote the input of BN within $m$ mini-batch data. BN standardizes and affine transforms the input by:

$$
\tilde{x} = \gamma\hat{x} + \beta = \gamma\frac{x - \mu}{\sigma} + \beta. \tag{2}
$$

Here $\tilde{x}$ denotes the output of BN. $\hat{x}$ denotes the standardized input. $\gamma$ and $\beta$ are learnable scale and shift parameters to recover a possible reduced representation capacity. In this paper, we also refer to the scale parameter and shift parameter as affine parameters. $\mu$ and $\sigma^2$ are the mean and variance of the mini-batch during training and population statistics during inference.

Compared to other normalization methods, such as Layer Normalization (LN), Instance Normalization (IN), etc., BN is particularly well-suited for SNNs. First, BN uses population statistics during inference, thereby avoiding the floating-point multiplication and division operations required to compute statistics. Second, BN can be integrated with linear transformations. Assuming the input current is the result of a linear transformation, i.e., $x = \mathbf{wz} + b$, BN can be integrated with the linear transformation as:

$$
\tilde{x} = \gamma\frac{\mathbf{wz} + b - \mu}{\sigma} + \beta = \frac{\gamma}{\sigma}\mathbf{wz} + \left(\frac{b - \mu}{\sigma} + \beta\right). \tag{3}
$$

Therefore, BN does not impair the spike-driven nature of SNNs and incurs no additional computational overhead during inference, making it particularly suitable for SNNs.

### 3.3. Energy consumption model

Following (Shi et al., 2024a), we use the number of synaptic operations (SOPs) to estimate the energy consumption of SNNs as follows:

$$
E = C_E \cdot \#\text{SOP} = C_E \sum_i s_i c_i, \tag{4}
$$

where $\#\text{SOP}$ denotes the total number of synaptic operations. Similar to (Shi et al., 2024a), we define a synaptic operation as the operation performed when a spike is passed through a synaptic connection to a postsynaptic neuron. Therefore, the total number of synaptic operations can be calculated by summing the number of synaptic connections each spike passed through, i.e., $\#\text{SOP} = \sum_i s_i c_i$. Here $s_i$ denotes the number of spikes fired by the $i$-th neuron, and $c_i$ denotes the number of synaptic connections from the $i$-th neuron. $C_E$ denotes the energy consumption corresponding to one SOP. It's worth noting that $C_E$ is the energy consumption of the entire system corresponding to one SOP on average rather than an accumulation operation only. For a hardware-independent theoretical analysis, we treat $C_E$ as a known constant.

# 4. Method

In this section, we first formulate the optimization objective of the sparsification with the energy constraint. Then we introduce the spike sparsification and weight pruning methods, respectively. Finally, we introduce UniSparse, a unified sparsification framework integrating the weight pruning and spike sparsification methods. An overview of the proposed UniSparse is presented in Fig. 1.

## 4.1. Sparsification with energy constraints

Given a spiking neural network $f(\cdot)$ which maps sample $x$ to $f(x; \boldsymbol{w})$ with weights $\boldsymbol{w} \in \mathbb{R}^{d_w}$, we first formulate the learning of the sparse structure and spikes as a loss minimization problem with energy constraints as follows:

$$\arg\min_{\boldsymbol{w}} \mathcal{L}(f(\cdot; \boldsymbol{w})) + \lambda E(\cdot; \boldsymbol{w}, \boldsymbol{s}). \quad (5)$$

Here $\mathcal{L}(f(\cdot; \boldsymbol{w}))$ denotes the loss of the prediction of the model $f(\cdot; \boldsymbol{w})$, $E(\cdot; \boldsymbol{w}, \boldsymbol{s})$ denotes the energy consumption penalty term, $\lambda$ controls the trade-off between task performance and energy consumption. $\boldsymbol{w} \in \mathbb{R}^{d_w}$ is the weights of the model, $\boldsymbol{s} \in \{0, 1\}^{d_n T}$ is the outputs of the neurons in the model, $d_w$ and $d_n$ are the number of weights and neurons, respectively. $T$ is the simulation time steps. Based on Eq. (4), the energy consumption penalty term can be formulated as follows:

$$E(\cdot; \boldsymbol{w}, \boldsymbol{s}) = \sum_t^T \sum_i s_i[t] \sum_j \mathbf{1}\{w_{ij} \neq 0\}, \quad (6)$$

where $s_i[t]$ is the output of the $i$-th presynaptic neuron at time-step $t$, $w_{ij}$ is the weight of the $j$-th synaptic connection from the $i$-th presynaptic neuron. $\mathbf{1}\{\cdot\}$ is the indicator function whose value is 1 if and only if the condition is true and 0 otherwise. $C_E$ is omitted since it can be absorbed in $\lambda$.

According to Eq. (6), to reduce the energy consumption of the network $f(\cdot)$, a promising way is to reduce the number of spikes or non-zero weights. Therefore, in the following subsections, we propose a spike sparsification method and a weight pruning method to achieve the energy constraint.

For simplicity in the following text, we rewrite Eq. (6) in vector form as:

$$E(\cdot; \boldsymbol{s}) = \|\boldsymbol{s} \odot g_s(\boldsymbol{s})\|_1, \quad (7)$$
$$E(\cdot; \boldsymbol{w}) = \|g_w(\boldsymbol{w}) \odot \mathbf{1}\{\boldsymbol{w} \neq 0\}\|_1, \quad (8)$$

where $\odot$ is the element-wise product (Hadamard product). Here we define two maps $g_s(\cdot)$ and $g_w(\cdot)$, where $g_s(\cdot)$ maps a neuron output to the number of synaptic connections it will transmit through, $g_w(\cdot)$ maps a weight of synaptic connection to the number of spikes transmitted on this connection.

We convert the network structure information embedded in the multiple summations in Eq. (6) to these two mappings, thus simplifying it to vector form.

## 4.2. Sparsification of spikes

Combining Eq. (1) and Eq. (2), we unfold the membrane potential along the time dimension, and we restate the neural dynamics of the spiking neurons as follows:

$$s[t] = H\left(u_{\text{rest}} + X[t] - v_{\text{th}}\right), \quad (9)$$

$$X[t] = \sum_{i=t_p+1}^{t} \left(\frac{1}{\tau}(1 - \frac{1}{\tau})^{t-i}\left(\gamma\frac{x[i] - \mu}{\sigma} + \beta\right)\right). \quad (10)$$

Here $x[i]$ is the input of the neuron at time-step $i$ before normalization, $X[t]$ is the accumulation of input current after decay from the last time the neuron fired a spike. $t_p$ denotes the last time the neuron fired a spike. Let $\hat{x}[t] = (x[t] - \mu)/\sigma$, $\hat{x}[t]$ denotes the normalized input. We assume that $\hat{x}[t]$ is a random variable with mean 0 and variance 1. Consider the firing condition:

$$u_{\text{rest}} + \sum_{i=t_p+1}^{t} \left(\frac{1}{\tau}(1 - \frac{1}{\tau})^{t-i}\left(\gamma\hat{x}[t] + \beta\right)\right) \geq v_{\text{th}}. \quad (11)$$

Assuming, without loss of generality, that $\gamma > 0$, Eq. (11) can be rearranged as:

$$\sum_{i=t_p+1}^{t} \left(\frac{1}{\tau}(1 - \frac{1}{\tau})^{t-i}\hat{x}[t]\right) \geq \frac{v_{\text{th}} - u_{\text{rest}} - D(t)\beta}{\gamma}. \quad (12)$$

Here we abbreviate $D(t) = \sum_{i=t_p+1}^{t} \frac{1}{\tau}(1 - \frac{1}{\tau})^{t-i} = 1 - (1 - \frac{1}{\tau})^{t-t_p}$. Let $\hat{z}[t] = \sum_{i=t_p+1}^{t} \frac{1}{\tau}(1 - \frac{1}{\tau})^{t-i}\hat{x}[t]$, $\hat{z}[t]$ is also a random variable. Therefore, the firing rate, i.e., the probability that the neuron fires a spike should be:

$$P(s[t] = 1) = 1 - F_z\left(\frac{v_{\text{th}} - u_{\text{rest}} - D(t)\beta}{\gamma}\right), \quad (13)$$

where $F_z(\cdot)$ denotes the cumulative distribution function (CDF) of $\hat{z}[t]$. Eq. (13) demonstrates that the firing rate of neurons can be reduced by increasing the term $(v_{\text{th}} - u_{\text{rest}} - D(t)\beta)/\gamma$. In other words, the affine parameters act as the learnable threshold of spiking neurons. We summarize this as the following theorem:

**Theorem 4.1.** *Given a LIF neuron with membrane time constant $\tau > 1$, threshold $v_{\text{th}}$, resting potential $u_{\text{rest}}$, and $T$ simulation time steps for inference. Consider its preceding normalization layer, which has affine parameters $\gamma$ and $\beta$. Assume that the normalized input $\hat{x}[t] = (x[t] - \mu)/\sigma$ is a random variable with a mean of 0 and variance of 1. If $\gamma > 0$ and $\beta < (v_{\text{th}} - u_{\text{rest}})/(1 - (1 - 1/\tau)^T)$, then the expectation of the accumulation of output $E(\sum_{t=1}^{T} s[t])$ monotonically increases with $\gamma$ and $\beta$.*

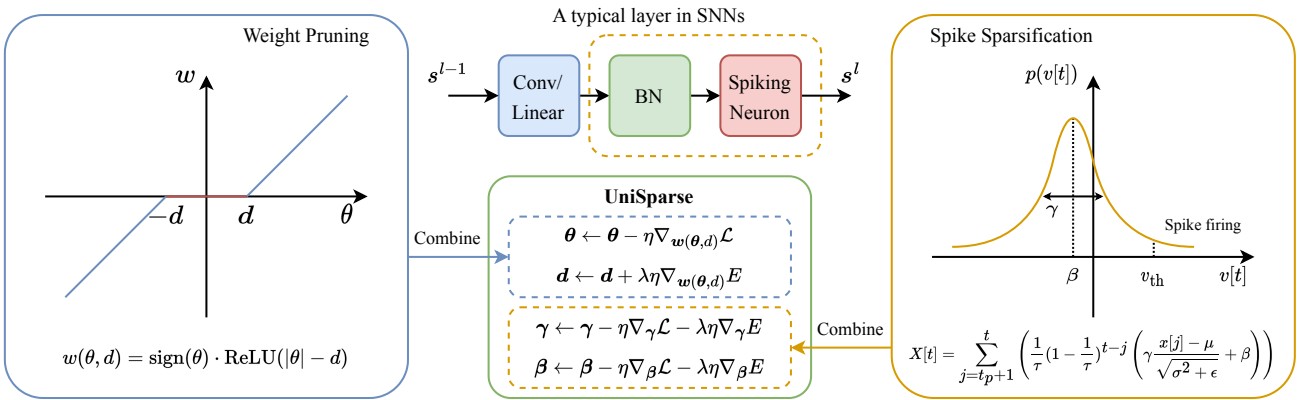

*Figure 1.* Overview of the proposed UniSparse

The proof can be found in the appendix. According to Theorem 4.1, it is promising to reduce spikes by constraining $\gamma$ and $\beta$. Eq. (13) also demonstrates why raising the threshold may not effectively reduce the firing rate, as the affine parameter can counteract this effect.

Based on the proof of effectiveness, we now derive the spike sparsification method. Considering the batch normalization layers, the original minimization objective in Eq. (5) with energy constraint in Eq. (7) can be reformulated as:

$$\underset{\boldsymbol{w},\boldsymbol{\gamma},\boldsymbol{\beta}}{\arg\min}\ \mathcal{L}(f(\cdot;\boldsymbol{w},\boldsymbol{\gamma},\boldsymbol{\beta})) + \lambda\|\boldsymbol{s}\odot g_s(\boldsymbol{s})\|_1. \tag{14}$$

The Gradient Descent (GD) update equation of $\boldsymbol{\gamma}$ and $\boldsymbol{\beta}$ should be:

$$\boldsymbol{\gamma} \leftarrow \boldsymbol{\gamma} - \eta\nabla_{\boldsymbol{\gamma}}\mathcal{L} - \lambda\eta\nabla_{\boldsymbol{\gamma}}\|\boldsymbol{s}\odot g_s(\boldsymbol{s})\|_1, \tag{15}$$

$$\boldsymbol{\beta} \leftarrow \boldsymbol{\beta} - \eta\nabla_{\boldsymbol{\beta}}\mathcal{L} - \lambda\eta\nabla_{\boldsymbol{\beta}}\|\boldsymbol{s}\odot g_s(\boldsymbol{s})\|_1. \tag{16}$$

Here we abbreviate $\mathcal{L}(f(\cdot;\boldsymbol{w},\boldsymbol{\gamma},\boldsymbol{\beta}))$ as $\mathcal{L}$. $\eta$ denotes the learning rate. To calculate the gradient of energy constraint $\|\boldsymbol{s}\odot g_s(\boldsymbol{s})\|_1$ with respect to $\boldsymbol{\gamma}$ and $\boldsymbol{\beta}$, we first calculate the partial derivative of $s[t]$ with respect to $\gamma$ and $\beta$ according to Eq. (9) and Eq. (10):

$$\begin{aligned}\frac{\partial s[t]}{\partial\gamma} &= \Theta'(v[t]-v_{\text{th}})\hat{z}[t],\\ \frac{\partial s[t]}{\partial\beta} &= \Theta'(v[t]-v_{\text{th}})D(t),\end{aligned} \tag{17}$$

where $\Theta(\cdot)$ is the surrogate function. Since the Heaviside function $H(\cdot)$ is non-derivable, following (Neftci et al., 2019), we use the derivative of the surrogate function $\Theta(x) = \frac{1}{\pi}\arctan(\pi x) + \frac{1}{2}$ as an approximation. $\Theta'(x) = 1/(1+(\pi x)^2)$ is the derivative of the surrogate function.

Since we consider the normalization with the spiking neuron as a whole, we only calculate the gradient of the energy constraint of the neuron output $\boldsymbol{s}^{(l)}$ of layer $l$ with respect to

the $\boldsymbol{\gamma}^{(l)}$ and $\boldsymbol{\beta}^{(l)}$ of its immediately preceding normalization layer. Therefore, combining Eq. (15), Eq. (16), and Eq. (17), the update equations should be:

$$\boldsymbol{\gamma}^{(l)} \leftarrow \boldsymbol{\gamma}^{(l)} - \eta\nabla_{\boldsymbol{\gamma}^{(l)}}\mathcal{L} - \lambda\eta g_s(\boldsymbol{s}^{(l)})h(\Theta'^{(l)}\hat{\boldsymbol{z}}^{(l)}), \tag{18}$$

$$\boldsymbol{\beta}^{(l)} \leftarrow \boldsymbol{\beta}^{(l)} - \eta\nabla_{\boldsymbol{\beta}^{(l)}}\mathcal{L} - \lambda\eta g_s(\boldsymbol{s}^{(l)})h(\Theta'^{(l)}D^{(l)}). \tag{19}$$

Here we abbreviate the surrogate gradient as $\Theta'^{(l)}$. Since the affine parameters are shared by all neurons in the layer, we define an aggregation function $h(\cdot)$ to simplify representation. $h(\cdot)$ sums the gradients of the spike outputs with respect to the affine parameters across all neurons in the layer, i.e., $h(\Theta'^{(l)}\hat{\boldsymbol{z}}^{(l)}) = \frac{1}{N^{(l)}}\sum_{t=1}^{T}\sum_{i=1}^{N^{(l)}}\Theta'(v_i^{(l)}[t] - v_{\text{th}})\hat{z}_i^{(l)}[t]$.

### 4.3. Pruning of weights

Inspired by (Kusupati et al., 2020; Chen et al., 2022), we use soft threshold reparameterization for weight pruning. It employs the soft threshold operator (Donoho, 2002) and optimizes over the reparameterized weights to introduce sparsity to the weights. The reparameterized weights are defined as:

$$w(\theta, d) = \text{sign}(\theta)\cdot\text{ReLU}(|\theta| - d), \tag{20}$$

where $w(\cdot,\cdot)$ is the original weight, $\theta$ is the reparameterization weight, $d \in \mathbb{R}^+$ is the soft threshold. $\text{sign}(\cdot)$ is the sign function. $\text{ReLU}(x) = \max(x, 0)$. Eq. (20) indicates that if $|\theta| \leq d$, the reparameterized weight $w(\theta, d) = 0$, thus introducing sparsity to weights. Different from (Kusupati et al., 2020), here $d$ is non-learnable but will update in each training iteration to control the sparsity. Furthermore, unlike the heuristic approach to scheduling $d$ in (Chen et al., 2022), we derive the update equation for $d$ from energy constraints, thereby automatically scheduling $d$ based on the connection patterns of each layer.

With the reparameterization in Eq. (20), the original minimization objective in Eq. (5) with energy constraint in

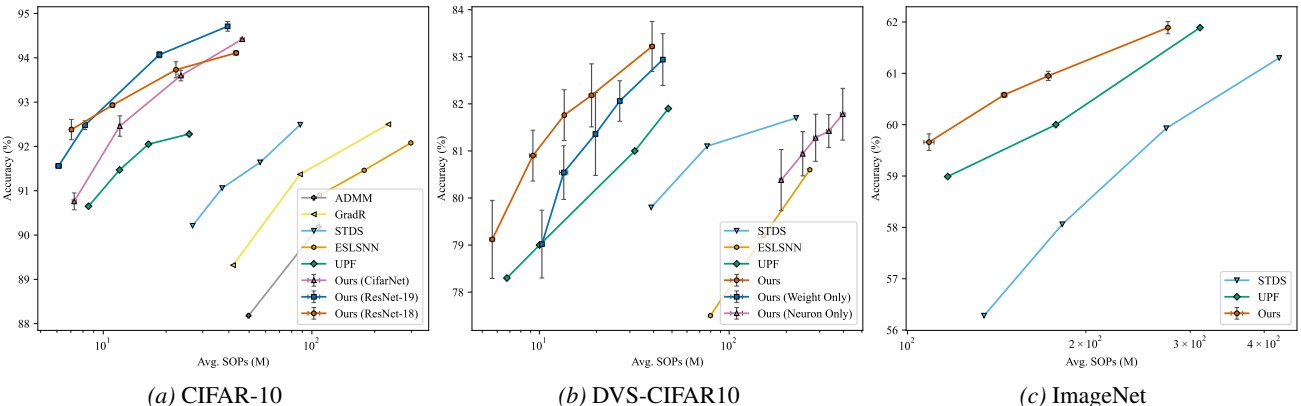

*(a)* CIFAR-10      *(b)* DVS-CIFAR10      *(c)* ImageNet

*Figure 2.* Comparison on different datasets

Eq. (8) can be reformulated as:

$$\arg\min_{\boldsymbol{\theta}} \mathcal{L}(f(\cdot; \boldsymbol{w}(\boldsymbol{\theta}, \boldsymbol{d}))) \\ + \lambda \|g_w(\boldsymbol{w}(\boldsymbol{\theta}, \boldsymbol{d})) \odot \mathbf{1}\{\boldsymbol{w}(\boldsymbol{\theta}, \boldsymbol{d}) \neq 0\}\|_1. \tag{21}$$

Here $\mathbf{1}\{\boldsymbol{w}(\boldsymbol{\theta}, \boldsymbol{d}) \neq 0\}$ is non-differentiable. Moreover, $\|\mathbf{1}\{\boldsymbol{w}(\boldsymbol{\theta}, \boldsymbol{d}) \neq 0\}\|_1$ is equal to the $L_0$ norm $\|\boldsymbol{w}(\boldsymbol{\theta}, \boldsymbol{d})\|_0$, whose convex relaxation $L_1$ norm $\|\boldsymbol{w}(\boldsymbol{\theta}, \boldsymbol{d})\|_1$ is differentiable almost everywhere. Therefore, we use the convex relaxation as an alternative and rewrite Eq. (21) as:

$$\arg\min_{\boldsymbol{\theta}} \mathcal{L}(f(\cdot; \boldsymbol{w}(\boldsymbol{\theta}, \boldsymbol{d}))) + \lambda\|g_w(\boldsymbol{w}(\boldsymbol{\theta}, \boldsymbol{d})) \odot \boldsymbol{w}(\boldsymbol{\theta}, \boldsymbol{d})\|_1. \tag{22}$$

With the minimization objective in Eq. (22), we first formulate the hypothetical Gradient Descent (GD) update equation of original weights as follows:

$$\boldsymbol{w} \leftarrow \boldsymbol{w} - \eta\nabla_{\boldsymbol{w}(\boldsymbol{\theta}, \boldsymbol{d})}\mathcal{L} - \lambda\eta g_w(\boldsymbol{w}(\boldsymbol{\theta}, \boldsymbol{d})) \odot \operatorname{sign}(\boldsymbol{w}(\boldsymbol{\theta}, \boldsymbol{d})). \tag{23}$$

Here we abbreviate $\mathcal{L}(f(\cdot; \boldsymbol{w}(\boldsymbol{\theta}, \boldsymbol{d})))$ as $\mathcal{L}$. $\nabla_{\boldsymbol{w}(\boldsymbol{\theta}, \boldsymbol{d})}\mathcal{L}$ is the gradient of loss $\mathcal{L}$ with respect to weight $\boldsymbol{w}(\boldsymbol{\theta}, \boldsymbol{d})$. Directly applying Eq. (23) may cause the weights to oscillate near 0 since we use $L_1$ norm relaxation. $L_1$ norm will reduce the absolute value of each weight $w$ in $\boldsymbol{w}$. However, when the magnitude of $w$ is small, the $L_1$ norm may result in $w$ oscillating around 0 instead of converging precisely to 0. To address this, we use an alternative approach to constrain the absolute value of $w$. We notice that a decrease in the absolute value $w$ is equivalent to an increase in the threshold $d$. Consequently, we convert the penalty on weights to an increase of threshold $d$ to avoid the oscillation. We formulate the update equations of reparameterization weights and threshold as:

$$\boldsymbol{\theta} \leftarrow \boldsymbol{\theta} - \eta\nabla_{\boldsymbol{w}(\boldsymbol{\theta}, \boldsymbol{d})}\mathcal{L} \odot \nabla_{\boldsymbol{\theta}}\boldsymbol{w}(\boldsymbol{\theta}, \boldsymbol{d}), \tag{24}$$
$$\boldsymbol{d} \leftarrow \boldsymbol{d} + \lambda\eta g_w(\boldsymbol{w}(\boldsymbol{\theta}, \boldsymbol{d})). \tag{25}$$

According to Eq. (20), the derivative $\mathrm{d}w/\mathrm{d}\theta = 0$ if $|\theta| \leq d$. Therefore, according to Eq. (24), the reparameterization

weight $\theta$ will no longer be updated when falling into the interval $[-d, d]$, and the weight $w$ will be constant at zero. However, a regrowth mechanism for reviving pruned weights is necessary, as pointed out in (Chen et al., 2021; 2022). Consequently, we set $\mathrm{d}w/\mathrm{d}\theta \equiv 1$ for pruned weights following (Chen et al., 2022), which is proved to be efficient and converges. The final update equations should be:

$$\boldsymbol{\theta} \leftarrow \boldsymbol{\theta} - \eta\nabla_{\boldsymbol{w}(\boldsymbol{\theta}, \boldsymbol{d})}\mathcal{L}, \tag{26}$$
$$\boldsymbol{d} \leftarrow \boldsymbol{d} + \lambda\eta g_w(\boldsymbol{w}(\boldsymbol{\theta}, \boldsymbol{d})). \tag{27}$$

### 4.4. Unified sparsification framework

In Sec. 4.2 and Sec. 4.3, we have introduced the spike sparsification method and the weight pruning method, respectively. Both methods use a uniform energy constraint and thus can be naturally integrated. Therefore, we propose UniSparse, a unified sparsification framework that combines both methods. The combined minimization objective can be formulated as:

$$\arg\min_{\boldsymbol{\theta}, \boldsymbol{\gamma}, \boldsymbol{\beta}} \mathcal{L}(f(\cdot; \boldsymbol{w}(\boldsymbol{\theta}, \boldsymbol{d}), \boldsymbol{\gamma}, \boldsymbol{\beta})) + \lambda E(\cdot; \boldsymbol{w}(\boldsymbol{\theta}, \boldsymbol{d}), \boldsymbol{s}). \tag{28}$$

We notice that the minimization objective has the same form as Eq. (14) and Eq. (21), i.e., the loss is the same, and the energy constraints are derived from the same energy consumption model. Therefore, the GD update equations for the combined minimization objective should be the combination of Eq. (18), Eq. (19), Eq. (26), and Eq. (27).

According to Eq. (27), the pruning strength for a weight depends on the number of spikes transmitted. If spikes are suppressed, fewer spikes are transmitted, leading to weaker pruning strength and thus denser synaptic connections. However, according to Eq. (18) and Eq. (19), denser synaptic connections in turn increase the strength of spike sparsification, which further suppresses spike activity. Similarly, sparser weights lead to stronger pruning pressure, resulting in further weight pruning.

*Table 1.* Comparison with the state-of-the-art methods on CIFAR-10

| Dataset | Method | Type | Architecture | T | Accuracy (%) | Avg. SOPs (M) |
|---------|--------|------|--------------|---|--------------|---------------|
| CIFAR-10 | ADMM (Deng et al., 2021) | Weight Pruning & Spike Sparsification | 7 Conv, 2 FC | 8 | 90.19
88.18 | 107.97
49.72 |
| | Grad R (Chen et al., 2021) | Weight Pruning | CifarNet | 8 | 92.50
91.37
89.32 | 232.51
87.73
41.89 |
| | STDS (Chen et al., 2022) | Weight Pruning | CifarNet | 8 | 92.49
91.64
91.06
90.21 | 87.94
56.49
37.16
26.81 |
| | ESLSNN (Shen et al., 2023) | Weight Pruning | ResNet-19 | 2 | 92.08
91.46
90.90 | 298.24
178.10
108.89 |
| | UPF (Shi et al., 2024a) | Weight Pruning & Neuron Pruning | CifarNet | 8 | 92.63
92.05
91.47
90.65 | 38.32
16.47
11.98
8.50 |
| | **UniSparse (This work)** | Weight Pruning & Spike Sparsification | CifarNet | 4 | **94.42**±0.01
**93.60**±0.12
**92.46**±0.23
**90.76**±0.19 | **46.37**±1.01
**23.56**±0.71
**12.01**±0.18
**7.26**±0.16 |
| | | | ResNet-19 | 4 | **94.71**±0.11
**94.07**±0.07
**92.48**±0.10
**91.56**±0.04 | **39.47**±0.38
**18.58**±0.48
**8.18**±0.14
**6.11**±0.20 |
| | | | ResNet-18 | 4 | **94.11**±0.05
**93.73**±0.18
**92.93**±0.05
**92.38**±0.23 | **43.30**±1.29
**22.31**±0.20
**11.07**±0.29
**7.04**±0.14 |

This positive feedback causes the training to degrade easily, meaning synaptic connections may be excessively pruned or neurons may be overly suppressed, especially when $\lambda$ is large. Therefore, similar to (Shi et al., 2024a), we use constants that describe the initial network structure information $g_w^{(l)}$ and $g_s^{(l)}$ to approximate the maps $g_w(\cdot)$ and $g_s(\cdot)$. Here $g_w^{(l)}$ denotes the number of presynaptic neurons in a feature map in the $l$-th layer, $g_s^{(l)}$ denotes the number of synaptic connections from a presynaptic neuron in the $l$-th layer. Based on the approximation, the GD update equation should be:

$$\boldsymbol{\theta}^{(l)} \leftarrow \boldsymbol{\theta}^{(l)} - \eta \nabla_{\boldsymbol{w}^{(l)}(\boldsymbol{\theta}^{(l)}, d^{(l)})} \mathcal{L}, \qquad (29)$$

$$d^{(l)} \leftarrow d^{(l)} + \lambda \eta g_w^{(l)}, \qquad (30)$$

$$\boldsymbol{\gamma}^{(l)} \leftarrow \boldsymbol{\gamma}^{(l)} - \eta \nabla_{\boldsymbol{\gamma}^{(l)}} \mathcal{L} - \lambda \eta g_s^{(l)} h(\Theta'^{(l)} \hat{\boldsymbol{z}}^{(l)}), \qquad (31)$$

$$\boldsymbol{\beta}^{(l)} \leftarrow \boldsymbol{\beta}^{(l)} - \eta \nabla_{\boldsymbol{\beta}^{(l)}} \mathcal{L} - \lambda \eta g_s^{(l)} h(\Theta'^{(l)} D^{(l)}). \qquad (32)$$

# 5. Experiments

In this section, we evaluate the effectiveness of the proposed UniSparse. We first evaluate its performance on various classification tasks, including CIFAR-10 (Krizhevsky et al., 2009), DVS-CIFAR10 (Li et al., 2017), and ImageNet (Deng et al., 2009) datasets using various SNN architectures. We also compare UniSparse with state-of-the-art methods. We focus on the accuracy and energy consumption of the sparsified model at different sparsity levels. Then, we perform an ablation study employing either the proposed weight pruning method or the spike sparsification method alone. Detailed experimental settings can be found in Appendix.

## 5.1. Comparison with the state-of-the-art methods

For each dataset, we conduct multiple sets of experiments to evaluate the accuracy and energy consumption of the sparsified model at different sparsity levels. In particular, we use three different architectures, CifarNet, ResNet-19, and ResNet-18 on the CIFAR-10 dataset to verify the effectiveness of the proposed UniSparse to sparsify different architectures. We compare the results with the state-of-the-art methods. Experimental results are listed in Tab. 1, Tab. 2, and Tab 3. For ease of comparison, we visualize the results in Fig. 2a, Fig. 2b, and Fig. 2c. The x-axis represents average SOPs in logarithmic coordinates, and the y-axis represents accuracy. Data points closer to the upper left indicate higher accuracy under the same energy consumption, which means better performance of the sparsification

*Table 2.* Comparison with the state-of-the-art methods on DVS-CIFAR10

| Dataset | Method | Type | Architecture | T | Accuracy (%) | Avg. SOPs (M) |
|---|---|---|---|---|---|---|
| DVS-CIFAR10 | STDS (Chen et al., 2022) | Weight Pruning | VGGSNN | 10 | 81.7
81.1
79.8 | 225.34
76.56
38.85 |
| | ESLSNN (Shen et al., 2023) | Weight Pruning | VGGSNN | 10 | 80.6
79.2
77.5 | 266.17
152.64
79.65 |
| | UPF (Shi et al., 2024a) | Weight Pruning &
Neuron Pruning | VGGSNN | 10 | 81.9
81.0
79.0
78.3 | 47.81
31.86
10.02
6.75 |
| | **UniSparse
(This work)** | Weight Pruning &
Spike Sparsification | VGGSNN | 10 | **83.22**±0.53
**82.18**±0.67
**81.76**±0.54
**80.90**±0.54
**79.12**±0.83 | **39.26**±1.19
**18.84**±0.56
**13.53**±0.23
**9.25**±0.37
**5.65**±0.17 |

*Table 3.* Comparison with the state-of-the-art methods on ImageNet

| Dataset | Method | Type | Architecture | T | Accuracy (%) | Avg. SOPs (M) |
|---|---|---|---|---|---|---|
| ImageNet | STDS (Chen et al., 2022) | Weight Pruning | ResNet-18 | 4 | 61.30
59.93
58.06
56.28 | 424.07
273.39
182.49
134.64 |
| | UPF (Shi et al., 2024a) | Weight Pruning &
Neuron Pruning | ResNet-18 | 4 | 61.89
60.00
58.99 | 311.70
177.99
116.88 |
| | **UniSparse
(This work)** | Weight Pruning &
Spike Sparsification | ResNet-18 | 4 | **61.89**±0.12
**60.95**±0.09
**60.58**±0.05
**59.66**±0.16 | **275.17**±0.48
**172.83**±2.08
**145.58**±1.65
**108.72**±2.15 |
| | | | Spikformer-8-384 | 4 | **72.21**
**70.68**
**68.29** | **2576.85**
**1717.47**
**1138.52** |

method. UniSparse outperforms state-of-the-art methods across various datasets.

Specifically, on the CIFAR-10 dataset, UniSparse achieves 90.76% accuracy with 7.26M SOPs using CifarNet, 91.56% accuracy with 6.11M SOPs using ResNet-19, and 92.38% accuracy with 7.04M SOPs using ResNet-18. All of these results have higher accuracy and fewer SOPs compared to the current leading method UPF, which achieves 90.65% accuracy with 8.50M SOPs. Similarly, on the DVS-CIFAR10 dataset, UniSparse achieves 79.12% accuracy with 5.65M SOPs, outperforming UPF 0.82% accuracy with 1.1M fewer SOPs. On the ImageNet dataset, UniSparse achieves 59.66% accuracy with 108.72M SOPs, outperforming UPF 0.67% accuracy with 8.16M fewer SOPs. Moreover, we also conduct experiments using Spikformer (Zhou et al., 2023), which is a spiking transformer architecture. Results in Tab. 3 show that UniSparse also performs well on spiking transformer, indicating its superior generalization capabilities.

### 5.2. Ablation study

To validate the effectiveness of combining the weight pruning method and the spike sparsification method, we conduct experiments where we only adopt the weight pruning method or the spike sparsification method and compare their performance with the proposed UniSparse on DVS-CIFAR10. For a fair comparison, all experimental settings follow the setup on DVS-CIFAR10. We add the data points to Fig. 2b. As shown in Fig. 2b, the proposed framework UniSparse outperforms the separate weight pruning method and spike sparsification method. Only adopting the spike sparsification method cannot reduce the SOPs to a low level, since it does not fundamentally eliminate the redundancy in deep SNNs as the redundant parameters remain. As shown in Fig. 2b, the data points using only the spike sparsification method are far to the lower right of the UniSparse. However, when incorporated with the weight pruning method, it helps to further reduce the SOPs without impairing the performance of the model. As shown in Fig. 2b, the data points of UniSparse are at the upper left of the weight pruning only.

This demonstrates the effectiveness of combining weight pruning and spike sparsification.

## 6. Conclusion

In this paper, we propose a spike sparsification method, which reduces spikes by constraining the affine parameters in normalization, and a weight pruning method based on soft threshold reparameterization. Combining both methods, we propose UniSparse, a unified sparsification framework for enhancing the energy efficiency of SNNs. Experimental results demonstrate that UniSparse achieves a state-of-the-art trade-off between accuracy and energy efficiency. Our work highlights the great potential of deep SNNs in improving energy efficiency.

## Acknowledgements

This work is supported by the National Natural Science Foundation of China (U24B20140, 62422601), Beijing Municipal Science and Technology Program (Z251100008125052), and Qiyuan Innovative Talent Program.

## Impact Statement

This paper presents work whose goal is to advance the field of Machine Learning. There are many potential societal consequences of our work, none which we feel must be specifically highlighted here.

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

## A. Proof of Theorem 4.1

*Proof.* We use a proof by mathematical induction. We first prove that the theorem is true when $T = 1$. The expectation $E(\sum_{t=1}^{T} s[t]) = E(s[1])$ when $T = 1$. We formulate the expectation as:

$$
\begin{aligned}
E(s[1]) &= 0 \cdot P(s[1] = 0) + 1 \cdot P(s[1] = 1) \\
&= P(s[1] = 1) \\
&= P\left(u_{\text{rest}} + \frac{1}{\tau}(\gamma z[1] + \beta) - v_{\text{th}} \geq 0\right) \\
&= P\left(\frac{1}{\tau} z[1] \geq \frac{v_{\text{th}} - u_{\text{rest}} - \frac{1}{\tau}\beta}{\gamma}\right).
\end{aligned}
\tag{33}
$$

Thus, $E(s[1])$ monotonically decreases with term $(v_{\text{th}} - u_{\text{rest}} - \beta/\tau)/\gamma$. Since $\gamma > 0$ and $\beta < (v_{\text{th}} - u_{\text{rest}})/(1 - (1 - 1/\tau)^T) = \tau \cdot (v_{\text{th}} - u_{\text{rest}})$, the expectation monotonically increases with $\gamma$ and $\beta$, which proves that the theorem is true when $T = 1$.

For the inductive step, we prove that for each $k \in \mathbb{N}$, if the theorem is true when $T = k$, then the theorem is true when $T = k + 1$. We notice that the expectation can be decomposed into

$$
E\left(\sum_{t=1}^{k+1} s[t]\right) = E\left(\sum_{t=1}^{k} s[t] + s[k+1]\right) = E\left(\sum_{t=1}^{k} s[t]\right) + E(s[k+1]).
\tag{34}
$$

Since the theorem is true when $T = k$ and $\beta < (v_{\text{th}} - u_{\text{rest}})/(1 - (1 - 1/\tau)^{k+1}) < (v_{\text{th}} - u_{\text{rest}})/(1 - (1 - 1/\tau)^{k})$, $E\left(\sum_{t=1}^{k} s[t]\right)$ monotonically increases with $\gamma$ and $\beta$. For $E(s[k+1])$, we have

$$
\begin{aligned}
E(s[k+1]) &= P\left(u_{\text{rest}} + \sum_{j=t_p+1}^{k+1}\left(\frac{1}{\tau}(1 - \frac{1}{\tau})^{k+1-j}(\gamma z[t] + \beta)\right) - v_{\text{th}} \geq 0\right) \\
&= P\left(\sum_{j=t_p+1}^{k+1}\left(\frac{1}{\tau}(1 - \frac{1}{\tau})^{k+1-j} z[t]\right) \geq \frac{v_{\text{th}} - u_{\text{rest}} - (1 - (1 - \frac{1}{\tau})^{k+1-t_p})\beta}{\gamma}\right).
\end{aligned}
\tag{35}
$$

Thus, $E(s[k+1])$ monotonically decreases with term $(v_{\text{th}} - u_{\text{rest}} - (1 - (1 - 1/\tau)^{k+1-t_p})\beta)/\gamma$. We notice that $1 \leq t_p \leq k$, therefore $k + 1 - t_p \leq k$. Since $\gamma > 0$ and $\beta < (v_{\text{th}} - u_{\text{rest}})/(1 - (1 - 1/\tau)^T) < (v_{\text{th}} - u_{\text{rest}})/(1 - (1 - 1/\tau)^{k})$, the expectation monotonically increases with $\gamma$ and $\beta$, which proves that the theorem is true when $T = k + 1$. Thus, by the principle of mathematical induction, the theorem is true for each $T \in \mathbb{N}^+$.

$\square$

Theorem 4.1 considers only the case that $\gamma > 0$. For the case that $\gamma < 0$, the monotonicity of $\beta$ is reversed. The proof is similar and will not be repeated here.

## B. Training of UniSparse

The training process of UniSparse using stochastic gradient descent (SGD) is summarized in Algorithm 1. The training process follows Eq. (29), Eq. (30), Eq. (31), and Eq. (32) in the main text. As illustrated in Algorithm 1, we first initialize the parameters $\boldsymbol{\theta}, \boldsymbol{\gamma}, \boldsymbol{\beta}$ following the initialization of typical SNNs, and set soft thresholds to 0. Then we perform standard training procedure. The difference is that we enlarge the soft threshold $d^{(l)}$ of each layer $l$ in each iteration according to the structure information $g_w^{(l)}$, which indicates the number of presynaptic neurons in a feature map in the $l$-th layer. And we add a penalization term to the parameters of batch normalization $\gamma^{(l)}$ and $\beta^{(l)}$ of each layer $l$ according to the structure information $g_s^{(l)}$, which indicates the number of synaptic connections from a presynaptic neuron in the $l$-th layer.

It should be noted that this algorithm does not introduce any additional learnable parameters. The only hyperparameter in this algorithm is $\lambda$, which controls the level of sparsity. Thus, UniSparse is easy to apply on existing methods without modifying training settings.

---

**Algorithm 1** UniSparse

---

**Input:** Network $f$, reparameterization weights $\boldsymbol{\theta}$, soft threshold $\boldsymbol{d}$, parameters $\boldsymbol{\gamma}, \boldsymbol{\beta}$, hyperparameter $\lambda$, learning rate $\eta$, training epochs $N$

**Output:** Sparsified network $f(\cdot; \boldsymbol{w}(\boldsymbol{\theta}, \boldsymbol{d}), \boldsymbol{\gamma}, \boldsymbol{\beta})$

1: Calculate network structure information $g_w^{(l)}$ and $g_s^{(l)}$ for each layer $l$;
2: Initialize parameters $\boldsymbol{\theta}, \boldsymbol{\gamma}, \boldsymbol{\beta}$;
3: Initialize soft threshold $\boldsymbol{d} = \boldsymbol{0}$;
4: **for** epoch $i = 0$ to $N$ **do**
5:    **for** batch $x$ in training set **do**
6:       **for** layer $l$ in network $f$ **do**
7:          $\boldsymbol{\theta}^{(l)} \leftarrow \boldsymbol{\theta}^{(l)} - \eta \nabla_{\boldsymbol{w}^{(l)}(\boldsymbol{\theta}^{(l)}, d^{(l)})} \mathcal{L}$
8:          $d^{(l)} \leftarrow d^{(l)} + \lambda \eta g_w^{(l)}$
9:          $\boldsymbol{\gamma}^{(l)} \leftarrow \boldsymbol{\gamma}^{(l)} - \eta \nabla_{\boldsymbol{\gamma}^{(l)}} \mathcal{L} - \lambda \eta g_s^{(l)} h(\Theta'^{(l)} \hat{\boldsymbol{z}}^{(l)})$
10:         $\boldsymbol{\beta}^{(l)} \leftarrow \boldsymbol{\beta}^{(l)} - \eta \nabla_{\boldsymbol{\beta}^{(l)}} \mathcal{L} - \lambda \eta g_s^{(l)} h(\Theta'^{(l)} D^{(l)})$
11:       **end for**
12:    **end for**
13: **end for**

---

## C. Experimental Settings

**CIFAR-10**   CIFAR-10 (Krizhevsky et al., 2009) is a static image dataset. It contains 60,000 RGB images with a $32 \times 32$ size. All the images are categorized into 10 classes with 6,000 samples in each class, including 5,000 training samples and 1,000 test samples. We normalize the input such that the samples have a mean of 0 and a variance of 1. For data augmentation, we adopt random horizontal flipping, random cropping with a padding of 4, and AutoAugment (Cubuk et al., 2019) to avoid over-fitting.

We use CifarNet, ResNet-19, and ResNet-18 architectures for CIFAR-10 classification. CifarNet is a convolutional SNN architecture with 6 convolution layers and 2 fully-connected layers ([[256C3]×3-MP2]×2-FC×2), also referred to as 6 Conv, 2 FC in (Chen et al., 2021; 2022; Shi et al., 2024a). ResNet-19 (Zheng et al., 2021) is a variant of the ResNet architecture. It has only 3 stages with 2 downsample layers. The ResNet-18 used in CIFAR-10 experiments is modified to fit the CIFAR-10 dataset. It replaces the $7 \times 7$ convolution and maxpooling with a $3 \times 3$ convolution. Both ResNet-19 and ResNet-18 use spiking-element-wise add (Fang et al., 2021) for shortcut connection.

All of our experiments on CIFAR-10 dataset follow the same training setup. We adopt SGD with 0.9 momentum optimizer with an initial learning rate of 0.1 and cosine decay to 0. We train the models for 300 epochs. The batch size and simulation time step are 256 and 4, respectively. For each sparsity level, we conduct 3 experiments with different random seeds to evaluate the random error. We report the mean and standard deviation of accuracy and average SOPs, and we report the ratio and accuracy loss based on the mean values.

**DVS-CIFAR10**   DVS-CIFAR10 (Li et al., 2017) is a neuromorphic dataset. It is created by regularly moving the images in CIFAR-10 dataset and capturing the movement using a dynamic vision sensor, thereby providing temporal information. It consists of 10,000 event streams. All the event streams are categorized into 10 classes with 1,000 samples in each class. Each sample has a spatial size of $128 \times 128$. Since DVS-CIFAR10 does not divide the samples into training and test sets, we select the first 900 samples of each category as training samples and the last 100 samples as test samples.

We adopt the following preprocessing procedure. First, we split the event stream of each sample into 10 slices. Each slice is a contiguous segment of the event stream, which consists 1/10 events of the sample. Then, for each slice, we stack all of its events into a single frame with 2 channels, representing positive events and negative events respectively. Finally, we use this frame as the input for a time step. Consequently, the simulation step used on DVS-CIFAR10 dataset is 10. We resize the spatial size of the input to $48 \times 48$. For data augmentation, we adopt the data augmentation technique proposed in (Li et al., 2022).

We use the VGGSNN (Deng et al., 2022) architecture for DVS-CIFAR10 classification. VGGSNN is a convolutional SNN architecture with 8 convolution layers and 1 fully-connected layer (64C3-128C3-AP2-[256C3]×2-AP2-[[512C3]×2-

*Table 4.* Detailed experimental results

| Dataset | Architecture | $\lambda$ | Accuracy (%) | Acc. Loss (%) | Avg. SOPs (M) | Ratio | Density (%) | Firing Rate (%) |
|---|---|---|---|---|---|---|---|---|
| CIFAR-10 | CifarNet | (dense) | 94.42±0.17 | 0 | 388.44±6.79 | 1 | 100 | 5.09±0.08 |
| | | $2 \times 10^{-7}$ | 94.42±0.01 | +0.00 | 46.37±1.01 | 8.38 | 72.52±0.01 | 4.73±0.11 |
| | | $5 \times 10^{-7}$ | 93.60±0.12 | -0.82 | 23.56±0.71 | 16.49 | 45.92±0.02 | 3.61±0.10 |
| | | $1 \times 10^{-6}$ | 92.46±0.23 | -1.96 | 12.01±0.18 | 32.34 | 19.15±0.01 | 2.66±0.01 |
| | | $1.5 \times 10^{-6}$ | 90.76±0.19 | -3.66 | 7.26±0.16 | 53.50 | 8.84±0.05 | 2.14±0.07 |
| | | $2 \times 10^{-6}$ | 89.25±0.49 | -5.17 | 4.83±0.31 | 80.42 | 5.24±0.04 | 1.74±0.15 |
| | ResNet-19 | (dense) | 95.16±0.03 | 0 | 923.01±16.69 | 1 | 100 | 6.83±0.10 |
| | | $5 \times 10^{-7}$ | 94.71±0.11 | -0.45 | 39.47±0.38 | 23.39 | 7.29±0.03 | 5.52±0.07 |
| | | $1 \times 10^{-6}$ | 94.07±0.07 | -1.09 | 18.58±0.48 | 49.68 | 3.41±0.03 | 3.91±0.15 |
| | | $2 \times 10^{-6}$ | 92.48±0.10 | -2.68 | 8.18±0.14 | 112.84 | 1.52±0.01 | 2.28±0.04 |
| | | $2.5 \times 10^{-6}$ | 91.56±0.04 | -3.60 | 6.11±0.20 | 151.07 | 1.15±0.02 | 1.90±0.09 |
| | | $3 \times 10^{-6}$ | 90.61±0.10 | -4.55 | 4.83±0.10 | 191.10 | 0.91±0.01 | 1.61±0.05 |
| | ResNet-18 | (dense) | 94.43±0.15 | 0 | 262.03±4.68 | 1 | 100 | 10.39±0.20 |
| | | $5 \times 10^{-7}$ | 94.11±0.05 | -0.32 | 43.30±1.29 | 6.05 | 32.69±0.20 | 9.99±0.32 |
| | | $1 \times 10^{-6}$ | 93.73±0.18 | -0.70 | 22.31±0.20 | 11.74 | 15.14±0.24 | 8.65±0.06 |
| | | $2 \times 10^{-6}$ | 92.93±0.05 | -1.50 | 11.07±0.29 | 23.67 | 6.40±0.27 | 6.58±0.41 |
| | | $3 \times 10^{-6}$ | 92.38±0.23 | -2.05 | 7.04±0.14 | 37.22 | 3.88±0.27 | 5.15±0.03 |
| | | $4 \times 10^{-6}$ | 91.04±0.27 | -3.39 | 4.94±0.16 | 53.04 | 2.67±0.11 | 3.98±0.09 |
| DVS-CIFAR10 | VGGSNN | (dense) | 83.50±0.48 | 0 | 646.25±11.63 | 1 | 100 | 5.38±0.17 |
| | | $2 \times 10^{-6}$ | 83.22±0.53 | -0.28 | 39.26±1.19 | 16.46 | 11.44±0.04 | 4.69±0.05 |
| | | $5 \times 10^{-6}$ | 82.18±0.67 | -1.32 | 18.84±0.56 | 34.30 | 5.91±0.06 | 3.28±0.05 |
| | | $7 \times 10^{-6}$ | 81.76±0.54 | -1.76 | 13.53±0.23 | 47.76 | 4.49±0.04 | 2.80±0.10 |
| | | $1 \times 10^{-5}$ | 80.90±0.54 | -2.60 | 9.25±0.37 | 69.86 | 3.20±0.04 | 2.34±0.05 |
| | | $1.5 \times 10^{-5}$ | 79.12±0.83 | -4.38 | 5.65±0.17 | 114.38 | 1.96±0.03 | 1.76±0.09 |
| ImageNet | ResNet-18 | (dense) | 63.78±0.04 | 0 | 1545.91±14.31 | 1 | 100 | 15.37±0.27 |
| | | $7 \times 10^{-8}$ | 61.89±0.12 | -1.89 | 275.17±0.48 | 5.62 | 41.67±0.01 | 13.38±0.32 |
| | | $1.5 \times 10^{-7}$ | 60.95±0.09 | -2.83 | 172.83±2.08 | 8.94 | 29.86±0.06 | 11.29±0.21 |
| | | $2 \times 10^{-7}$ | 60.58±0.05 | -3.20 | 145.58±1.65 | 10.62 | 25.75±0.02 | 11.09±0.03 |
| | | $3 \times 10^{-7}$ | 59.66±0.16 | -4.12 | 108.72±2.15 | 14.22 | 20.75±0.06 | 9.50±0.60 |
| | Spikformer-8-384 | (dense) | 72.97 | 0 | 7856.82 | 1 | 100 | 14.61 |
| | | $1 \times 10^{-6}$ | 72.21 | -0.76 | 2576.85 | 3.05 | 52.24 | 11.15 |
| | | $2 \times 10^{-6}$ | 70.68 | -2.29 | 1717.47 | 4.57 | 39.27 | 13.51 |
| | | $3 \times 10^{-6}$ | 68.29 | -4.68 | 1138.52 | 6.90 | 33.07 | 11.38 |

[*] Accuracy, Avg. SOPs, Density, and Firing Rate on CIFAR-10 and DVS-CIFAR10 datasets are reported in mean±std form. We conduct experiments using different random seeds for 3 times on CIFAR-10 dataset and 5 times on DVS-CIFAR10 dataset.

AP2]×2-FC). The experiments on DVS-CIFAR10 dataset basically follow the setup in (Shi et al., 2024a). We adopt SGD with 0.9 momentum optimizer with an initial learning rate of 0.025 and cosine decay to 0. We train the models for 300 epochs. The batch size and simulation step are 64 and 4, respectively. Due to the significant random error on DVS-CIFAR10 dataset, we conduct 5 experiments for each sparsity level with different random seeds. Moreover, we adopt the TET (Deng et al., 2022) loss for experiments on DVS-CIFAR10 dataset.

**ImageNet** ImageNet (Deng et al., 2009) is a large-scale static image dataset. It is one of the most commonly used datasets in computer vision tasks. It consists of around 1.2 million high-resolution images, categorized into 1,000 classes. Each category contains approximately 1,000 images representing a wide variety of objects and scenes to effectively reflect real-world scenarios. We adopt the standard preprocessing procedure, i.e., data normalization, randomly crop and resize the images to 224×224, and random horizontal flipping for data augmentation. We use the ResNet-18 architecture with spiking-element-wise add (Fang et al., 2021) shortcut connection for ImageNet classification.

For ResNet-18 architecture, we adopt SGD with 0.9 momentum optimizer with an initial learning rate of 0.1 and cosine decay to 0. We add a weight decay of $1 \times 10^{-4}$ to avoid over-fitting. We train the models for 300 epochs. The batch size is 256, and the simulation time step is 4. For Spikformer architecture, we adopt AdamW optimizer with an initial learning rate of 0.001 and cosine decay to 0. We adopt random augment, label smoothing of 0.1, random cutout, and weight decay of

0.05 to avoid over-fitting. We train the models for 300 epochs. The batch size is 512, and the simulation time step is 4. Due to the time-consuming training process on ImageNet dataset, we conduct only one experiment for each sparsity level.

## D. Detailed Experimental Results

Tab. 4 lists the detailed experimental results, including density and average firing rate under more sparsity levels. Here density denotes the number of remaining parameters as a percentage of the total number of parameters. For CIFAR-10 dataset, we add results of $\lambda = 2 \times 10^{-6}$ for CifarNet, $\lambda = 3 \times 10^{-6}$ for ResNet-19, and $\lambda = 4 \times 10^{-6}$ for ResNet-18. As listed in Tab. 4, the proposed UniSparse effectively reduces the density and firing rates.

