# OpenReview forum: "UniSparse: Combining Weight Pruning and Spike Sparsification in Spiking Neural Networks"
_ICML.cc/2026/Conference — ICML 2026 regular_

### Official Review · Reviewer_CKaf · 2026-03-10

**Soundness:** 3
**Presentation:** 3
**Significance:** 3
**Originality:** 3
**Overall Recommendation:** 5
**Confidence:** 4

**Summary:**

This work proposes a method to jointly maximize the weight and spike sparsity during the training of spiking neural networks. By adjusting hyperparameters, the method achieves a series of accuracy-energy trade-offs on several visual benchmarks.

**Compliance With Llm Reviewing Policy:**

Affirmed.

**Final Justification:**

My concerns have been adequately addressed. I raise my score.

**Key Questions For Authors:**

See Weaknesses.

**Limitations:**

See Weaknesses.

**Strengths And Weaknesses:**

Strengths:
1. This work is well-organized and easy to follow.
2. The paper provides both a theoretical foundation and publicly available source code, which is highly commendable.

Weaknesses:
1. The relationship between the theoretical analysis and the real implementation is unclear to me. If I understand correctly, once a loss function like Equation 7 is defined, the impact of various parameters (such as the affine parameters in Batch Normalization) on the loss will be automatically defined by the gradient (or surrogate gradient), and sparsification will be directly accomplished via autograd. In this context, what is the reason for emphasizing Batch Normalization and providing Theorem 4.1? Moreover, could Equation 7 be interpreted as the complete method for the spike sparsification part (it does indeed differ from other approaches)?

2. Based on the experimental results, the performance of the proposed sparsification/pruning method appears to be suboptimal. Specifically, on the most challenging task in the paper—Spikformer on ImageNet—a 1/2 reduction in energy consumption leads to a 4% drop in accuracy. This raises concerns about the algorithm's scalability, and a more comprehensive evaluation is required to fully assess its strengths and limitations.

3. Although the SOP metric is frequently used in the meuromorphic computing area, it is mainly used to measure the performance of a chip. It is not a quite direct metric for measuring energy, which is related to both computation and memory access.

Minor:
1. Formulas should be presented in accordance with academic standards (e.g., each formula should be followed by a comma or a period).

---

> ### Author Rebuttal · Authors · 2026-03-30
>
> We appreciate the time and effort you invested in reviewing our paper. We will address all the concerns you have raised.
>
> > Weakness 1: Relationship between theoretical analysis and implementation.
>
> Thank you for pointing this out. The difference between the implementation of the proposed spike sparsification method and the standard autograd process lies in the gradient propagation. Specifically, we compute the gradient of the energy constraint term for the output spikes $\boldsymbol{s}^{(l)}$ of layer $l$ **only** with respect to the affine parameters $\boldsymbol{\gamma}^{(l)}$ and $\boldsymbol{\beta}^{(l)}$ of its immediately preceding normalization layer. By truncating the gradient flow from the spike energy constraint to earlier layers, we decouple weight pruning and spike sparsification. This ensures that the spike energy constraint influences only the affine parameters, without interfering with the weight pruning objective. Theorem 4.1 formally demonstrates the effectiveness of reducing spike activity through constraints on these affine parameters.
>
> > Weaknesses 2: Performance drop on Spikformer with ImageNet.
>
> The results presented in Table 3 of the main text report accuracy and SOPs of models after applying sparsification. We acknowledge that further increasing sparsity yields diminishing reductions in SOPs. For a more comprehensive comparison, Table 4 in the appendix provides detailed results, including the accuracy and SOPs of the dense (unpruned) model. As shown in Table 4, UniSparse reduces SOPs by 85.5% (corresponding to 6.72G SOPs) compared to the dense model, with an accuracy drop of 4.68%. This demonstrates that UniSparse remains effective for spiking vision transformers, achieving significant energy savings while maintaining competitive performance.
>
> > Weaknesses 3: Suitability of SOPs as an energy efficiency metric.
>
> We agree that energy consumption depends on both computation and memory access. However, the energy consumption of computation and memory access on neuromorphic chips is influenced by specific hardware architecture designs. Consequently, for hardware-independent analysis, we use Synaptic Operations (SOPs) as a proxy metric for energy consumption. The Synaptic Operations (SOPs) metric used in this paper refers to the total number of synaptic operations required during inference, which is hardware-independent. This differs from Synaptic Operations Per Second (SOPS), which is typically used to measure neuromorphic chip throughput.
>
> Using SOPs as a proxy for energy efficiency aligns with established practice in recent SNN research, enabling consistent and meaningful comparisons across studies. Prior works such as UPF [1], ESL-SNNs [2], Spikformer [3], and other modern architectures [4,5] have adopted SOPs to evaluate or estimate energy efficiency. Therefore, we believe SOPs provide a valid and widely accepted basis for assessing the energy-saving potential of sparsification methods.
>
> > Minor: Formulas should be presented in accordance with academic standards.
>
> Thank you for pointing this out! We will carefully review and revise all mathematical expressions to ensure they comply with academic formatting standards in the final version.
>
> References
>
> [1] Shi, X., et al. (2024). Towards energy efficient spiking neural networks: An unstructured pruning framework. ICLR.
>
> [2] Shen, J., et al. (2023). ESL-SNNs: An evolutionary structure learning strategy for spiking neural networks. AAAI.
>
> [3] Zhou, Z., et al. (2023). Spikformer: When Spiking Neural Network Meets Transformer. ICLR.
>
> [4] Hu, Y., et al. (2024). Advancing spiking neural networks toward deep residual learning. TNNLS.
>
> [5] Yao, M., et al. (2025). Scaling spike-driven transformer with efficient spike firing approximation training. TPAMI.

---

> > ### Author Rebuttal · Reviewer_CKaf · 2026-04-01
> >
> > All of my concerns have been resolved. I recommend integrating these clarifying details into the final revision to enhance the paper's overall clarity.

---

> > > ### Author Response · Authors · 2026-04-06
> > >
> > > Thank you for your thorough review and valuable feedback on our manuscript. We are delighted that our rebuttal has adequately addressed your concerns. We sincerely appreciate your recognition of our work and the constructive comments you provided, which have greatly enhanced the paper's quality. We will integrate these clarifying details into the final revision.

---

### Official Review · Reviewer_69NE · 2026-03-12

**Soundness:** 3
**Presentation:** 3
**Significance:** 3
**Originality:** 3
**Overall Recommendation:** 5
**Confidence:** 3

**Summary:**

This paper presents UniSparse, a unified framework for improving the energy efficiency of SNNs by jointly performing weight pruning and spike sparsification under a common energy constraint. The key theoretical contribution is the observation that Batch Normalization affine parameters ($\gamma$ and $\beta$) effectively serve as learnable firing thresholds for subsequent spiking neurons (Theorem 4.1). Based on this, the authors propose reducing firing rates by constraining BN affine parameters via an energy-derived gradient term, selectively suppressing less important spikes. For weight pruning, they adopt soft threshold reparameterization where the threshold $d$ is automatically scheduled from the same energy constraint rather than through heuristic tuning. Both methods share a single hyperparameter $\lambda$ controlling the overall sparsity. Experiments on CIFAR-10, DVS-CIFAR10, and ImageNet with CNNs and a spiking transformer show that UniSparse achieves favorable accuracy vs. energy (SOPs) tradeoffs compared to prior methods.

**Compliance With Llm Reviewing Policy:**

Affirmed.

**Final Justification:**

The author has provided a well-structured and effective solution that thoroughly addresses the issue I encountered. The approach is both clear and practical. Accordingly, I am raising my score by one point.

**Key Questions For Authors:**

Do all layers end up with similar weight density and firing rates, or does the method naturally allocate more sparsity to certain layers? A layer-by-layer breakdown of density and firing rate at convergence (e.g., for ResNet-19 on CIFAR-10) would help assess whether the uniform $\lambda$ constraint leads to reasonable layer-wise behavior or if layer-adaptive $\lambda$ could improve results.

Tables 1-3 compare methods at their reported settings, but different methods achieve different accuracy-SOPs tradeoff curves. A fairer comparison would fix a target SOP budget and compare accuracy, or fix target accuracy and compare SOPs. Could you provide such a matched comparison, at least against UPF? This would make the claimed improvements much more convincing.

**Limitations:**

yes

**Strengths And Weaknesses:**

Energy efficiency is the primary motivation for deploying SNNs on neuromorphic hardware. Achieving 37x compression on ResNet-18 (CIFAR-10) with only about 2% accuracy drop and 151x compression on ResNet-19 is practically meaningful. The framework introduces no additional learnable parameters, and the single $\lambda$ hyperparameter keeps the method easy to use.

The energy model based on synaptic operations (SOPs) is well-established in the SNN literature. The experimental protocol is reasonable: multiple runs with standard deviations on CIFAR-10 and DVS-CIFAR10, standard architectures, and comparison with recent baselines (UPF 2024, STDS 2022, ESLSNN 2023).

Weaknesses:
1. The approximations $g_w^{(l)}$ and $g_s^{(l)}$ used as layer-wise constants simplify the energy model significantly. In reality, the energy contribution of different neurons and weights varies depending on activation patterns. This constant approximation means all neurons in a layer receive the same sparsification pressure regardless of their actual contribution to energy consumption. The paper does not discuss whether this leads to suboptimal allocation of sparsity across layers.

2. On ImageNet (Table 3), only single runs are reported due to computational cost. Without error bars, it is difficult to assess whether UniSparse's improvement over UPF (61.90% vs 61.89% on ResNet-18) is meaningful or simply within noise. The Spikformer result (72.21%) is reported without a comparison at the same sparsity level against other methods, making it hard to contextualize.

---

> ### Author Rebuttal · Authors · 2026-03-30
>
> Thank you for your time and effort invested in reviewing our paper. We will address all the questions you have raised.
>
> > Weakness 1: The constant approximation of $g_w^{(l)}$ and $g_s^{(l)}$.
>
> Thank you for pointing this out. While using the true values of $g_w$ and $g_s$ could better capture the actual connection and activation patterns of the network, this also tends to make the optimization problem ill-posed, as noted in prior work UPF.
>
> According to Equation (27), the pruning strength for a weight depends on the number of spikes transmitted. If spikes are suppressed, fewer spikes are transmitted, leading to weaker pruning strength and thus denser synaptic connections. However, according to Equations (18) and (19), denser synaptic connections in turn increase the strength of spike sparsification, which further suppresses spike activity. Similarly, sparser weights lead to stronger pruning pressure, resulting in further weight pruning.
>
> This positive feedback causes the training to degrade easily, meaning synaptic connections may be excessively pruned or neurons may be overly suppressed, especially when $\lambda$ is large. Therefore, similar to UPF, we use constants that describe the initial network structure information $g_w^{(l)}$ and $g_s^{(l)}$ to approximate the maps $g_w$ and $g_s$. This helps stabilize the optimization process.
>
> We will include this analysis in the final version.
>
> > Weakness 2: Experimental evaluation on ImageNet and comparison with other methods.
>
> Thank you for pointing this out. We conduct two additional runs with different random seeds for each sparsity level. Results are listed in Table R5 in the form of mean$\pm$std.
>
> *Table R5. Results on ImageNet.*
>
> | $\lambda$          | Accuracy (%)   | Avg. SOPs (M)   |
> | ------------------ | -------------- | --------------- |
> | $7\times10^{-8}$   | 61.89$\pm$0.12 | 275.17$\pm$0.48 |
> | $1.5\times10^{-7}$ | 60.95$\pm$0.09 | 172.83$\pm$2.08 |
> | $2\times10^{-7}$   | 60.58$\pm$0.05 | 145.58$\pm$1.65 |
> | $3\times10^{-7}$   | 59.66$\pm$0.16 | 108.72$\pm$2.15 |
>
> As shown in Table 3 of the main text and Table R5, UniSparse achieves accuracy comparable to UPF (61.89% vs. 61.89%) while significantly reducing SOPs (275.40M vs. 311.70M). This demonstrates that UniSparse offers a better trade-off between performance and energy efficiency.
>
> Regarding comparisons on spiking vision transformer architectures, to the best of our knowledge, existing methods have not reported results on such backbones. We are currently evaluating these methods on Spikformer through our own implementation. However, due to time constraints of rebuttal and the substantial training time required for Spikformer on ImageNet, we have not yet completed these experiments. We will include these results in the final version.
>
> > Question 1: Layer-wise breakdown of density and firing rate.
>
> Thank you for your suggestion. Due to space constraints in rebuttal, we are unable to list the density and firing rates for every layer of ResNet-19. As an alternative, we provide these metrics for the final layer of each stage in Table R6.
>
> *Table R6. Density and firing rates for the final layer of each stage in ResNet-19 (*$\lambda=2\times10^{-6}$*).*
>
> | Stage | Density (%) | Firing Rate (%) |
> | ----- | ----------- | --------------- |
> | 1     | 0.15        | 0.62            |
> | 2     | 0.30        | 0.92            |
> | 3     | 1.66        | 1.30            |
>
> As shown in Table R6, UniSparse automatically assigns higher sparsity to earlier layers. This is because the synaptic weights in earlier layers connect to a larger number of presynaptic neurons, leading to higher sparsification strength. A similar phenomenon is observed in other architectures, indicating that UniSparse naturally induces layer-adaptive sparsity without requiring manual tuning of layer-specific hyperparameters.
>
> > Question 2: Fair comparison under fixed SOP budgets or target accuracy.
>
> Unfortunately, it is not feasible to directly control the SOPs of pruned SNNs. For a detailed explanation, please refer to our response to Reviewer rPyi Weakness 1. Similarly, it is also not possible to control the test accuracy of pruned SNNs directly. Therefore, following common practice as adopted by UPF, we compare methods using accuracy–SOPs trade-off curves. As shown in Figure 2 in the main text, for each data point of existing methods, there is a corresponding UniSparse data point located to its upper left. This indicates that UniSparse achieves higher accuracy with lower SOPs across varying sparsity levels, indicating a consistently superior trade-off between performance and energy efficiency.

---

> > ### Author Rebuttal · Reviewer_69NE · 2026-04-04
> >
> > The author has provided a well-structured and effective solution that thoroughly addresses the issue I encountered. The approach is both clear and practical. Accordingly, I am raising my score by one point.

---

> > > ### Author Response · Authors · 2026-04-06
> > >
> > > Thank you for your dedicated effort and thoughtful review of our manuscript. We are glad that our rebuttal has resolved the concerns you raised. We sincerely appreciate your recognition of our work and the valuable feedback you have provided. Your insightful comments have been instrumental in improving the quality and clarity of the paper. We will include these clarifications in the final revision.

---

### Official Review · Reviewer_aACz · 2026-03-13

**Soundness:** 2
**Presentation:** 2
**Significance:** 2
**Originality:** 2
**Overall Recommendation:** 4
**Confidence:** 3

**Summary:**

This paper proposes UniSparse, a unified sparsification framework for improving the energy efficiency of deep spiking neural networks by jointly reducing parameter redundancy and spike redundancy. The key observation is that, in SNNs with batch normalization before spiking neurons, the BN affine parameters can be interpreted as learnable thresholds that affect firing rates. Based on this, the paper introduces a spike sparsification method that constrains affine parameters to reduce spikes, and combines it with a soft-threshold-reparameterization-based weight pruning method derived from the same energy objective. Experiments on CIFAR-10, DVS-CIFAR10, and ImageNet across several SNN architectures show improved accuracy–SOP trade-offs over prior methods, with additional ablations suggesting that combining spike sparsification and weight pruning is more effective than either component alone.

**Compliance With Llm Reviewing Policy:**

Affirmed.

**Final Justification:**

Thanks for the rebuttal, my concern has been resolved, so increase my rating to 4.

**Key Questions For Authors:**

Please address concerns in Weakness section.

**Limitations:**

Yes

**Strengths And Weaknesses:**

Strength

[1] The problem setting is well motivated and clearly framed. The paper argues that improving the energy efficiency of deep SNNs requires addressing both parameter redundancy and spike redundancy, rather than treating them in isolation.

[2] The method appears practically lightweight. The paper emphasizes that UniSparse does not introduce additional learnable parameters and controls sparsity with a single hyperparameter, which is an attractive property from a usability and implementation standpoint


Weakness

[1] The related work positioning is somewhat oversimplified. The paper frames prior methods largely as addressing either parameter redundancy or spike redundancy, and suggests that unified frameworks are rare. However, the paper itself acknowledges prior joint approaches such as Deng et al. (2021) and Shi et al. (2024). This means the main novelty is not simply the existence of a joint optimization framework, but rather the specific way UniSparse instantiates that framewor. I therefore think the paper should position its novelty more precisely relative to these prior joint methods, rather than implying that the unified perspective itself is largely new.

[R1] Deng et al. (2021), Comprehensive SNN compression using ADMM optimization and activity regularization, IEEE TNNLS.
[R2] Shi et al. (2024), weight pruning + neuron pruning unified framework for SNN efficiency, cited by this paper as a key joint prior.


[2]  The ablation study is useful but still somewhat limited relative to the central claims. It does show that the joint method outperforms weight-only or spike-only variants, which is important. However, given that much of the paper’s novelty lies in the specific design of spike sparsification through BN affine constraints, I would have liked more detailed decomposition—for example, comparisons against simpler threshold-raising baselines, analysis of how the affine-parameter constraint behaves in layers without BN, or a clearer breakdown of which design choices matter most. This would make the claimed contribution more transparent.

[3]  The paper would also be stronger if it discussed or validated how the proposed method extends to more recent transformer-based SNN backbones. Recent SNN research has increasingly explored transformer-style architectures, including Spikformer, STAtten, and newer spiking-transformer variants, suggesting that strong-performing SNN models are no longer limited to CNN-style backbones. Since UniSparse relies heavily on BN-affine-based spike control, it is not yet clear how directly the method transfers to transformer-based SNNs, where the normalization, token mixing, and architectural structure may differ substantially. I would therefore encourage the authors to clarify whether the method is expected to apply to transformer-based SNNs, and ideally provide at least one validation on such a backbone or a discussion of the necessary modifications.

[R3] Zhou et al. / Spikformer: A Key Foundation Model for Spiking Neural Networks (transformer-based SNN line).
[R4] Lee et al. / Spiking Transformer with Spatial-Temporal Attention (STAtten).
[R5] Guo et al. / Spiking Transformer: Introducing Accurate Addition-Only Spiking Self-Attentio

---

> ### Author Rebuttal · Authors · 2026-03-30
>
> We sincerely appreciate your thorough review of our paper and the invaluable insights you provided. We are grateful for the opportunity to address the points you raised.
>
> > Weakness 1: The related work positioning is somewhat oversimplified.
>
> Thank you for your suggestion. We will further clarify the novelty of UniSparse relative to the prior joint methods. Compared to existing joint methods, UniSparse offers distinct advantages in being both lightweight and easy to implement. Specifically, it introduces no additional learnable parameters and is naturally integrated into the training process, thereby avoiding significant computational overhead. In contrast, UPF relies on learnable masks for pruning, which introduces a significant number of additional trainable parameters and leads to increased computation and memory costs. Similarly, ADMM also introduces auxiliary variables and requires a separate retraining phase for pruning.
>
> Furthermore, UniSparse introduces only one hyperparameter $\lambda$, which controls the overall sparsity of the pruned SNNs. This eliminates the need for complex hyperparameter tuning. In contrast, UPF requires carefully controlled annealing for the learnable masks during training and introduces multiple hyperparameters to control the annealing process. ADMM requires empirical tuning of penalty term weights for weight pruning and spike sparsification.
>
> We will revise the introduction section in the final version to better highlight these advantages and the novelty of our approach.
>
> > Weakness 2: The ablation study is useful but still somewhat limited relative to the central claims.
>
> Thank you for your suggestion. We will provide a more detailed decomposition of the proposed spike sparsification method. To verify the claim that raising the firing threshold of spiking neurons has limited effectiveness in reducing spikes, we conduct experiments for the threshold-raising baseline on CIFAR-10 with ResNet-18. The results are presented in Table R3.
>
> *Table R3. Results under different thresholds.*
>
> | Threshold | Accuracy (%) | Avg. SOPs (M) |
> | --------- | ------------ | ------------- |
> | 2.0       | 94.21        | 227.23        |
> | 3.0       | 93.78        | 202.81        |
> | 4.0       | 93.87        | 195.91        |
> | 5.0       | 93.67        | 193.32        |
>
> As shown in Table R3, while raising the threshold can reduce SOPs to some extent, further reductions become difficult as the threshold continues to increase. Specifically, when the threshold rises from 4.0 to 5.0, SOPs reduce by only 2.59M. This is because the interaction between normalization layers and spiking neurons counteracts the effect of threshold increases. As formulated in Equation (13) of the main text, increases in the affine parameters $\gamma$ and $\beta$ may counteract the reduction in firing rate achieved by raising the threshold.
>
> For comparison, we also evaluate the proposed spike sparsification method under the same hyperparameter settings. The results are listed in Table R4.
>
> *Table R4. Results of spike sparsification.*
>
> | $\lambda$        | Accuracy (%) | Avg. SOPs (M) |
> | ---------------- | ------------ | ------------- |
> | $5\times10^{-6}$ | 94.74        | 198.18        |
> | $1\times10^{-5}$ | 94.27        | 175.97        |
> | $2\times10^{-5}$ | 93.20        | 129.79        |
> | $3\times10^{-5}$ | 92.78        | 105.57        |
>
> Table R4 demonstrates that our spike sparsification method can continuously reduce SOPs as the hyperparameter $\lambda$ increases. Furthermore, it achieves a better trade-off between accuracy and SOPs compared to simply raising the threshold.
>
> Regarding the behavior of the affine-parameter constraint in layers without BN, please refer to our response to Reviewer rPyi Weakness 2. Specifically, for other normalization methods such as LN, the affine-parameter constraint remains applicable without modification. For layers without normalization, a straightforward approach is to introduce additional affine parameters to the linear or convolutional layers preceding the spiking neurons. These parameters can be merged into the transformations after training, as described in Equation (3). Experimental results confirm that UniSparse remains effective in enhancing the energy efficiency of SNNs without BN, demonstrating its general applicability.
>
> > Weakness 3: Extension to transformer-based SNN backbones.
>
> Thank you for your suggestion. We would like to point out that we have conducted experiments on ImageNet using Spikformer [R3], which is a spiking vision transformer backbone. The results are included in Table 3 of the main text and Table 4 in the appendix. As shown in these tables, UniSparse effectively improves the energy efficiency of spiking vision transformers, demonstrating its applicability to modern SNN backbones.

---

> > ### Author Rebuttal · Reviewer_aACz · 2026-04-02
> >
> > Thanks for the rebuttal, my concern has been resolved, so increase my rating to 4.

---

> > > ### Author Response · Authors · 2026-04-06
> > >
> > > Thank you for your time and thoughtful review of our manuscript. We are pleased that our rebuttal has addressed your concerns. We sincerely appreciate your recognition of our work and the constructive feedback you provided, which helps enhance the quality of our paper. We will integrate these revisions into the final revision.

---

### Official Review · Reviewer_rPyi · 2026-03-13

**Soundness:** 3
**Presentation:** 3
**Significance:** 3
**Originality:** 3
**Overall Recommendation:** 4
**Confidence:** 4

**Summary:**

This paper introduces UniSparse, a sparsification framework designed to enhance the energy efficiency of SNNs. It reveals that the affine parameters in the normalization layer function as the learnable threshold for the subsequent spiking neuron layer. Building on this insight, the paper proposes a spike sparsification method that reduces spike activity by constraining the learnable parameters in batch normalization. Concurrently, it introduces a weight pruning method based on soft threshold reparameterization. Both methods are derived from the same energy constraint, enabling their natural integration into a unified framework. Experiments across various datasets and network architectures demonstrate that UniSparse achieves a state-of-the-art balance between accuracy and energy efficiency.

**Compliance With Llm Reviewing Policy:**

Affirmed.

**Key Questions For Authors:**

See weakness.

**Limitations:**

No, see weakness.

**Strengths And Weaknesses:**

Strengths:
1. This study reveals that the affine parameters in the normalization layer can serve as the learnable threshold for the subsequent spiking neuron layer, offering a novel perspective on the relationship between spiking neurons and normalization.
2. The approach of reducing spikes by constraining learnable parameters in batch normalization is innovative.
3. Extensive experiments demonstrate that UniSparse effectively enhances the energy efficiency of SNNs, outperforming state-of-the-art methods across diverse benchmarks and SNN architectures.

Weaknesses:

1. The proposed UniSparse controls the target sparsity level through the empirical hyperparameter $\lambda$ rather than directly specifying the target SOPs.
2. The proposed spike sparsification technique relies on the BN layers preceding the spiking neurons. Can this approach be applied to architectures using other normalization methods like Layer Normalization, or to those without normalization?
3. UniSparse integrates the weight pruning and spike sparsification methods into a unified framework under the same energy constraint. The necessity of this joint sparsification framework is not clear. What are the advantages of this joint sparsification framework compared to performing weight pruning and spike sparsification sequentially?

---

> ### Author Rebuttal · Authors · 2026-03-30
>
> We sincerely appreciate your helpful feedback. We will address all the concerns you have raised.
>
> > Weakness 1: Direct control of SOPs.
>
> Unfortunately, it is not feasible to directly control the SOPs of pruned SNNs. This is because SOPs are a dynamic metric that varies across input samples, unlike the fixed FLOPs in traditional ANNs. In SNNs, synaptic operations are triggered by sparse spikes, and each input sample generates a distinct spike pattern, leading to sample-dependent SOPs. The SOPs reported in the paper represent the average over all test samples. During training, however, the exact distribution of test samples is unknown, making it impossible to precisely predict or control the SOPs for each test sample. Therefore, controlling SOPs directly through a predefined target is not practical. Instead, we adjust the overall sparsity level via the hyperparameter $\lambda$, which indirectly influences the average SOPs.
>
> > Weaknesses 2: Applicability to other normalization methods or architectures without normalization.
>
> Yes, the proposed UniSparse is also applicable to architectures with LayerNorm or without normalization. To validate this, we conduct experiments on CIFAR-10 using ResNet-18 with BNs replaced by LNs. Furthermore, since deep SNNs without normalization typically struggle to converge, we conduct experiments on MNIST using a shallow two-layer fully-connected network without normalization. To maintain consistency in the spike sparsification method, we add additional affine parameters to each linear layer. These parameters can be merged into the linear transformations after training, as formulated in Equation (3). Experimental results are listed in Tables R1 and R2, respectively.
>
> *Table R1. Results on CIFAR-10 using ResNet-18 with LN.*
>
> | $\lambda$        | Accuracy (%) | Avg. SOPs (M) |
> | ---------------- | ------------ | ------------- |
> | 0 (dense)        | 93.22        | 170.51        |
> | $1\times10^{-7}$ | 92.85        | 71.54         |
> | $2\times10^{-7}$ | 92.75        | 46.11         |
> | $3\times10^{-7}$ | 92.34        | 32.29         |
> | $5\times10^{-7}$ | 91.81        | 19.97         |
>
> *Table R2. Results on MNIST using a 2-layer fully-connected network without normalization.*
>
> | $\lambda$        | Accuracy (%) | Avg. SOPs (M) |
> | ---------------- | ------------ | ------------- |
> | 0 (dense)        | 92.54        | 2.53          |
> | $1\times10^{-5}$ | 92.95        | 0.98          |
> | $2\times10^{-5}$ | 91.75        | 0.58          |
> | $3\times10^{-5}$ | 90.56        | 0.41          |
>
> The experimental results confirm that UniSparse effectively improves the energy efficiency of SNNs with LayerNorm or without normalization, demonstrating its general applicability.
>
> > Weaknesses 3: Advantages of a joint sparsification framework over sequential methods.
>
> Thank you for raising this important question. The joint formulation of weight pruning and spike sparsification within a unified optimization framework offers two key advantages.
>
> Firstly, jointly formulating weight pruning and spike sparsification under the same optimization objective guarantees an optimal assignment of penalty weights between neurons and synaptic weights. If independent weight pruning and spike sparsification methods are applied sequentially, it is necessary to manually and empirically adjust the strength of each method. Such empirical settings are not guaranteed to be optimal. In contrast, UniSparse automatically assigns penalty weights to each synaptic weight and neuron based on network structure information, and adjusts the strength of weight pruning and spike sparsification through a unified hyperparameter $\lambda$.
>
> Secondly, the unified framework accomplishes both weight pruning and spike sparsification in a single training process, reducing training overhead compared to sequentially applying existing techniques.

---

> > ### Author Rebuttal · Reviewer_rPyi · 2026-04-03
> >
> > The rebuttal has addressed my concerns reasonably well. Since my original recommendation was already positive, I am inclined to maintain my current score.

---

> > > ### Author Response · Authors · 2026-04-06
> > >
> > > Thank you for your thoughtful review and valuable feedback on our manuscript. We sincerely appreciate the time and effort you have dedicated to evaluating our work. We are pleased that our rebuttal has addressed your concerns, and we are grateful for the opportunity to further clarify and strengthen our paper.
> > >
> > > In our rebuttal, we provided detailed responses to each of the weaknesses you raised. Specifically, we clarified why directly controlling SOPs is not feasible due to the dynamic nature of spike activity in SNNs, demonstrated the applicability of UniSparse to architectures using Layer Normalization or without normalization through additional experiments, and explained the advantages of our joint sparsification framework over sequential methods.
> > >
> > > To further validate the advantages of the joint sparsification framework over sequentially applying weight pruning and spike sparsification, we conduct experiments on CIFAR-10 with ResNet-18. Specifically, during the first half of the training process, we perform weight pruning and then fix the resulting sparse weight structure. During the second half of the training process, we perform spike sparsification. Results are listed in the following Table R7.
> > >
> > > *Table R7. Results of sequential weight pruning and spike sparsification.*
> > >
> > > | $\lambda$        | Accuracy (%) | Avg. SOPs (M) |
> > > | ---------------- | ------------ | ------------- |
> > > | $5\times10^{-7}$ | 93.92        | 55.69         |
> > > | $1\times10^{-6}$ | 93.62        | 29.30         |
> > > | $2\times10^{-6}$ | 92.61        | 13.79         |
> > > | $3\times10^{-6}$ | 91.73        | 8.71          |
> > >
> > > For ease of comparison, we have reposted the experimental results from the main text in the following Table R8.
> > >
> > > *Table R8. Results of UniSparse.*
> > >
> > > | $\lambda$        | Accuracy (%) | Avg. SOPs (M) |
> > > | ---------------- | ------------ | ------------- |
> > > | $5\times10^{-7}$ | 94.11        | 43.30         |
> > > | $1\times10^{-6}$ | 93.73        | 22.31         |
> > > | $2\times10^{-6}$ | 92.93        | 11.07         |
> > > | $3\times10^{-6}$ | 92.38        | 4.94          |
> > >
> > > Tables R7 and R8 demonstrate that UniSparse achieves a better trade-off between accuracy and SOPs compared to sequentially applying weight pruning and spike sparsification. These results further validate the advantages of the joint sparsification framework of UniSparse.
> > >
> > > Thank you once again for your valuable comments and for helping us enhance the quality of our research.

---

### Decision · Program_Chairs · 2026-04-30

**Decision:**

Accept (regular)

**Comment:**

This paper addresses the energy efficiency optimization of spiking neural networks by proposing UniSparse, a unified sparsification framework. It innovatively reveals that the affine parameters in normalization layers can serve as learnable thresholds for subsequent spiking neurons, and jointly realizes weight pruning and spike sparsification under a unified energy constraint, with a clear theoretical motivation and a concise, efficient method design.Extensive experiments are conducted on multiple datasets and classic SNN architectures, including spiking Transformers, and the proposed method achieves a better trade-off between accuracy and SOP energy consumption compared with existing approaches. The effectiveness of the method is further verified through detailed comparative experiments and ablation studies.All key concerns raised by the four reviewers regarding applicability, experimental fairness, and the connection between theory and implementation have been thoroughly and reasonably addressed by the authors with supplementary experimental evidence. Overall, this work is technically rigorous and solidly contributes, with high academic value and practical significance. We therefore recommend acceptance.